# Definitive Treatment of Femoral Shaft Fractures: Comparison between Anterograde Intramedullary Nailing and Monoaxial External Fixation

**DOI:** 10.3390/jcm8081119

**Published:** 2019-07-28

**Authors:** Gianluca Testa, Andrea Vescio, Domenico Costantino Aloj, Giacomo Papotto, Luigi Ferrarotto, Alessandro Massé, Giuseppe Sessa, Vito Pavone

**Affiliations:** 1Department of General Surgery and Medical Surgical Specialties, Section of Orthopaedics and Traumatologic Surgery, AOU Policlinico-Vittorio Emanuele, University of Catania, 95123 Catania, Italy; 2Department of Orthopaedics Surgery, Division of Muscular-Skeletal Traumatology, AOU Città della Salute, CTO Hospital, 10126 Turin, Italy; 3Department of Traumatology, PO Sant’Andrea, 13100 Vercelli, Italy

**Keywords:** femoral shaft fractures, polytrauma, anterograde intramedullary nailing, monoaxial external fixation

## Abstract

Background: Femoral shaft fractures result from high-energy trauma. Despite intramedullary nailing (IMN) representing the gold standard option of treatment, external fixation (EF) can be used temporarily for damage control or definitively. The purpose of this study is to compare two different options, anterograde IMN and monoaxial EF, for the treatment of femoral shaft fractures. Methods: Between January 2005 and December 2014, patients with femoral shaft fractures operated on in two centers were retrospectively evaluated and divided into two groups: the IMN group (*n* = 74), and the EF group (*n* = 73). For each group, sex; laterality; age; and AO classification type mean follow-up, mean union time, and complications were reported. Results: Both groups were found to have no statistical differences (*p* > 0.05) in sex, laterality, age, and AO classification types. In the IMN group the average surgery duration was 79.7 minutes (range 45–130). The average time for bone union was 26.9 weeks. Major complications occurred in 4 (5.4%) patients. In the EF group the average follow-up duration was 59.8 months (range 28–160). The average time for bone union was 24.0 weeks. Major complications occurred in 16 (21.9%) patients. Conclusions: IMN is the gold standard for definitive treatment of femoral shaft fractures. In patients with severe associated injuries, EF should be a good alternative.

## 1. Introduction

Femoral shaft fracture incidence is approximatively 0.01% and results from high-energy trauma, often associated with polytrauma, comminuted fractures, and open fractures [1,2]. In the last four decades, several treatment options have been used to treat femoral shaft fractures. Initially, the treatment was represented by various types of trans-skeletal traction; then, the use of plates and screws was introduced, still indicated in special conditions [3,4]. Nowadays, reduction and fixation with intramedullary nail, introduced by Groves in the United Kingdom and by Kuntcher in Germany [5], is the gold-standard treatment for femoral shaft fractures [1,2]. External fixation is not frequently performed to treat femoral diaphyseal fractures and there have been few studies performed on this topic, in which its use is mainly indicated to temporarily stabilize the fracture in patients who suffer polytrauma or open fractures [5]. In such instances external fixation can be used as a temporary treatment and subsequently converted into intramedullary nailing within two weeks after the trauma [6]. There are many conditions in which intramedullary nailing cannot be performed—for example, in polytrauma patients whose conditions require major surgical procedures. In these cases, some surgeons perform external fixation as a definitive treatment for femoral shaft fractures to avoid further surgical sessions, reducing complications rates and costs [7,8,9].

The purpose of the following clinical study is to compare two different options, namely, anterograde intramedullary nailing and monoaxial external fixation, for the treatment of femoral shaft fractures, performed in two different trauma centers.

## 2. Experimental Section

### 2.1. Eligibility Criteria

Patients with femoral shaft fractures treated between January 2005 and December 2014 were retrospectively evaluated and divided into two groups: the intramedullary nailing (IMN) group, who underwent reduction and fixation with anterograde intramedullary nailing at the orthopedic clinic of the University of Catania; and the EF group, i.e., patients who underwent reduction and fixation with a monoaxial external fixator at C.T.O. Hospital in Turin. Inclusion criteria were: closed unilateral fractures of the femoral shaft (AO type 32) and aged between 18 and 75 years. Spontaneous and bilateral fractures were ruled out.

For each group the following parameters were reported: sex, laterality, age, AO classification type, mean follow-up, details of treatment, surgical duration, weight-bearing time, mean union time.

### 2.2. Clinical and Radiographic Assessment

Patients were followed up 3, 6, and 12 months post-operatively and then yearly using an SF-36 clinical questionnaire and radiographic evaluation. Fractures were considered united in the absence of movement and pain on stress at the fracture site. Radiographic union was achieved in the presence of uniform and continuous ossification of the callus, with consolidation and development of trabeculae across the fracture site [6].

Major and minor complications of both groups were reported. Union time of more than 26 weeks in closed fractures was considered a delayed union. The diagnosis of non-union was made in the presence of abnormal movement at the fracture site at least 9 months after the injury and with no progressive signs of healing for at least 3 months, despite continuing treatment. Malunion was defined with one of the following criteria: shortening of more than 2.5 cm, angulation of more than 10°, or rotational malalignment of more than 5° [6].

### 2.3. Statistical Analysis

The IMN group and the EF group were statistically compared using Student’s *t*-test for quantitative data and the chi-squared test for qualitative data.

## 3. Results

### 3.1. Demographic Data

The IMN group was composed of 74 patients, (49 (66.2%) males and 25 (33.8%) females). The Mean average age was 42.2 ± 20.3 years (range 18–75 years). Fracture involved the right side in 41(55.4%) patients and the left in 33 (44.6%). All the fractures were caused by road accidents. Three (4.1%) patients had a polytrauma. In 28 (37.8%) cases the fracture occurred at the proximal third of the femoral shaft, in 40 (54.1%) cases at the middle third and in 7 (8.1%) at the distal third. Using the AO classification system 7 (9.5%) fractures were classified as 32-A1, 21 (28.4%) as 32-A2, 17 (23.0%) as 32-A3, 11 (14.9%) as 32-B1, 7 (9.5%) as 32-B2, 4 (5.4%) as 32-B3, 1 (1.4%) as 32-C1, 4 (5.4%) as 32-C2, and 2 (2.7%) as 32-C3.

The EF group was composed of 73 patients, (53 (72.6%) males and 20 (27.4%) females). The mean average age was 40.2 ± 16.5 years (range 18–73 years). Fracture involved the right side in 32 (43.8%) patients and the left in 41 (56.2%). All the fractures were caused by road accidents. Twelve (16.4%) patients had a polytrauma. In 13 (17.8%) cases the fracture occurred at the proximal third of the femoral shaft, in 47 (64.4%) cases at the middle third and in 13 (17.8%) at the distal third. Using the AO classification system 2 (2.7%) fractures were classified as 32-A1, 18 (24.7%) as 32-A2, 12 (16.4%) as 32-A3, 8 (11.0%) as 32-B1, 9 (12.3%) as 32-B2, 7 (9.6%) as 32-B3, 2 (2.7%) as 32-C1, 4 (5.5%) as 32-C2, and 11 (15.1%) as 32-C3 (Table 1 and Figure 1).

### 3.2. Details of Treatment

In the IMN group the average follow-up duration was 65.4 ± 25.7 months (range 28–168). The nail was proximally locked with a cephalic screw in 21 (28.4%) patients and with diaphyseal screws in the remaining 54 (71.6%). Of 3 (4.1%) polytrauma patients, 1 (1.4%) underwent immediate intramedullary nailing and two (2.7%) underwent temporary external fixation within 24 hours with subsequent conversion to intramedullary nailing after an average time of 9 days. In the remaining 71 (95.9%) cases the average time between the injury and the intramedullary nailing was 5.6 ± 2.8 days (range 2–20), with a skeletal proximal tibial traction being performed in all cases when the patients were admitted. The average surgery duration was 79.7 ± 21.7 min (range 45–130). Patients were mobilized with the aid of crutches approximately 15 days after the surgery with a gradual increase in weight-bearing. Postoperative weight-bearing was allowed at a mean time of 21.2 ± 7.4 days (range 15–45) (Table 2). Ten patients (13.5%) had nail removal at a mean time of 23.1 ± 9.2 months (range 9–36). The average time for bone union was 26.9 ± 10.9 weeks (range 19–83). Excluding cases of delayed union, 69 (93.2%) fractures united at a mean time of 23.6 ± 3.2 weeks (range 19–31). In five cases (6.8%) delayed union was reported with an average healing time of 53.9 ± 14.6 weeks (range 35–83) (Figure 2).

In the EF group the average follow-up duration was 59.8 ± 22.1 months (range 28–160). The Procallus Orthofix® external fixation system was used for all 73 patients. In 56 (76.7%) cases monoaxial external fixation was performed within 24 hours after the trauma and in the remaining 17 (23.3%) cases after a delay of 7.3 ± 6.1 days (range 3–20). Before definitive treatment, in 5 (6.8%) cases a temporary proximal tibial skeletal traction was applied and in 12 (16.4%) a temporary external fixation was performed. The average surgery duration was 53.8 ± 10.0 minutes (range 35–80). Patients were mobilized with the aid of crutches immediately after the surgery with a gradual increase in weight-bearing. Postoperative weight-bearing was allowed at a mean time of 21.8 ± 28.3 days (range 0–120) (Table 2). The external fixator was removed when radiography demonstrated continuous and uniform bridging callus at the fracture site at a mean time of 6.6 ± 2.9 months (range 4–24). The average time for bone union was 24.0 ± 12.1 weeks (range 13–102). Excluding cases of delayed union, 61 (83.6%) fractures united at a mean time of 20.9 ± 3.5 weeks (range 13–30). For 12 cases (16.4%), delayed union was reported with an average healing time of 53.4 ± 22.8 weeks (range 36–102) (Figure 2).

### 3.3. Clinical and Radiographic Assessment

In the IMN group, the average SF-36 score was 89.5 ± 15 (range 63–97). Major complications occurred in 4 (5.4%) patients: there was 1 (1.3%) case of septic nonunion, 2 (2.7%) cases of malunion, and 1 (1.3%) case of loss of reduction. Additional surgery was performed in 7 (9.4%) cases: 1 (1.3%) case with intramedullary nail breakage was treated with a new nail implant, 2 (2.7%) cases of septic nonunion were treated with a monoaxial external fixation, and in 1 (1.3%) case a cerclage wiring was implanted due to loss of reduction. Minor complications occurred to 7 (9.4%) patients: there were 3 (4.0%) distal screw ruptures and 4 (5.4%) cases of post-operative knee pain (Figure 3).

In the EF group, the average SF-36 score was 83.4 ± 21.5 (range 51–94). Major complications occurred in 16 (21.9%) patients: 2 (2.8%) cases of septic nonunion were treated with surgical debridement and circular external fixation, 1 (1.4%) re-fracture was successfully treated with a new monoaxial external fixation, and 4 (5.5%) cases of loss of reduction were treated with an external fixator resetting in operating theatre. Furthermore, 2 (2.8%) cases of malunion were observed, 1 case of 3 cm bone shortening and varus deformity treated with circular external fixation, and 1 case of posterior bowing and internal rotation of 20° treated with hexapod external fixation. Additional surgery was performed in 10 (13.7%) cases due to delayed union. Persistent knee stiffness was observed in 1 case (1.4%) and was treated with Judet’s arthromyolysis.

Ten patients (13.7%) suffered minor complications: 9 (12.3%) cases of pin tract infection were treated with accurate pin tract medications and antibiotic therapy and 1 (1.4%) case of a broken Schanz screw was successfully removed (Figure 3).

### 3.4. Statistical Findings

Statistical findings have been reported in Table 2 and Figure 1 and Figure 2.

## 4. Discussion

This study compared two definitive treatment options, namely, antegrade intramedullary nailing and monoaxial external fixation, performed in two different trauma centers. It has been necessary to statistically compare individual criteria, such as sex, age, and type of fracture, to demonstrate the two groups’ statistical homogeneity (*p* < 0.05). Treatment of polytrauma patients requires an early and rapid approach, depending on their general condition and associated injuries [10]. There is a wide range of benefits resulting from early reduction and fixation: decrease of fat embolism risk caused by fracture movements, pain relief, and the consequent reduction of analgesic drugs requirement, and in addition early fixation shortens hospitalization. Early mobilization of pulmonary function in surgical patients improves their respiratory function, increasing functional residual capacity, preventing atelectasis, and decreasing pulmonary shunt [11]. Rogers et al. [12] tried to determine the best time to perform definitive fixation on patients with femoral shaft fractures, encountering few differences concerning pulmonary morbidity and hospitalization between immediate surgery (within 24 hours after the trauma) and early surgery (24/72 hours after the trauma). However, their studies showed an increased risk of infections and pulmonary complications in patients treated more than 72 hours after the injury. In this series the EF group was treated earlier than the IMN group (*p* < 0.05). In fact, 86% of EF group patients underwent definitive surgery within 24 hours after the trauma, whereas 75% of IMN group patients underwent temporary, proximal tibial, skeletal traction, with a mean delay of 5–6 days in performing definitive surgery. Pape and Tscherne evidenced that early reamed intramedullary nailing may increase the risk of acute respiratory distress syndrome (ARDS) in patients who have suffered a thoracic trauma. Their studies showed that delayed surgery or other options (non-reamed intramedullary nailing or external fixation) offer greater advantages in patients with thoracic trauma, rather than early reamed intramedullary nailing [13].

Moreover, a shorter surgical time may decrease peri- and post-operative complications in patients with femoral diaphyseal fractures, especially in those who are in a critical state due to high energy trauma [14]. The average surgical duration was 80 minutes in the IMN group and 50 minutes in the EF group (*p* < 0.0001). In the literature [2,8], the average surgical duration was 60 minutes for intramedullary nailing and 40 minutes for monoaxial external fixation.

Concerning polytrauma patients, the principle of “damage control surgery” was applied in both groups. This principle consists of an early stabilization of unstable fractures, bleeding control, and management of intracranial or abdominal conditions. When patients’ conditions are stable, a definitive fracture treatment may be performed. In these cases of damage control, the definitive treatment is represented by the conversion of temporary external fixation into intramedullary nailing. This treatment regimen was applied in two patients from the IMN group, while in 12 cases from the EF group, a new external fixator followed temporary external fixation to quicken patients’ recovery and allow for treatment of other associated non-skeletal injuries.

Although post-operative weight-bearing was allowed at a mean time of 21 days in both groups, the median values were 24 days for the IMN group and 10 days for the EF group. An early weight-bearing is an advantage of external fixation, but in polytrauma patients, the average recovery time is influenced by associated cranial, thoracic, abdominal or bone lesions. Unequal post-operative weight-bearing may be a factor that influences different union times of the two groups [15,16]. In fact, even if there are no significant differences in union times between the two groups, considering only the patients healed before 30 weeks and excluding delayed unions, the EF group had a mean union time 4 weeks less than the IMF group (20 vs 24 weeks). Average union times were in accordance with data from the published literature [7,8,9]. The AAOS suggests allowing immediate weight-bearing also to patients with comminute fractures treated with intramedullary nailing or external fixation, delaying it when fixation plates and screws are used [16]. Considering SF-36, no statistical significances were reported between the two groups, indicating any conditioning of chosen fixation devices in quality of life.

The major discrepancy between the two groups was recorded in complications, with a higher rate in the EF group. In the published literature, the most frequent complications described in patients treated with monoaxial external fixation are pin-tract infection and knee stiffness [17], which can be avoided with better hygiene, antibiotic therapy, and knee joint mobilization [13]. In this study pin-tract infection was reported in 12% of patients, whereas knee stiffness was present only in 1 patient who was treated by performing Judet’s arthromyolisis [17]. Cases of delayed and aseptic union were treated by performing additional surgery, such as bone graft, resetting of external fixation, or circular external fixation [7]. In the EF group, loss of reduction was a frequent complication which occurred especially in patients with type B or C fractures according to AO classification and treated with a new reduction and resetting of the external fixator.

In the IMN group, delayed union was treated conservatively using shockwave therapy [18] or surgically by removing the nail [19]. Septic unions were almost equal in both groups and treated with bone resection, toilette, and stabilization with circular external fixation, according to principles of distraction osteogenesis [20].

## 5. Conclusions

The surgical treatment of femoral shaft fractures has several options to be considered. Intramedullary nailing is the gold standard, showing good outcomes with low rates of complications. Moreover, in patients with comminuted fractures or associated injuries, definitive external fixation has been proven to be an ideal method for definitive fixation, because of minimal invasiveness, decreased blood loss and thromboembolism risk, earlier weight-bearing, and shorter hospitalization. Nevertheless, external fixation is associated with increased frequency complications; consequently, patient compliance and continuous clinical and radiological follow-up are imperative.

## Figures and Tables

**Figure 1 jcm-08-01119-f001:**
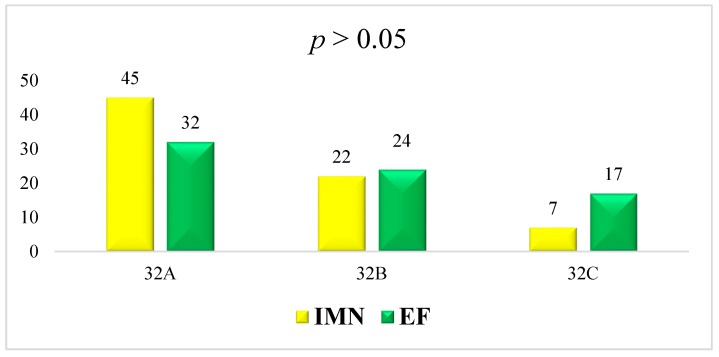
AO classification of IMN and EF groups. Both groups have no statistical differences (*p* > 0.05) in sex, laterality, age, and AO classification types.

**Figure 2 jcm-08-01119-f002:**
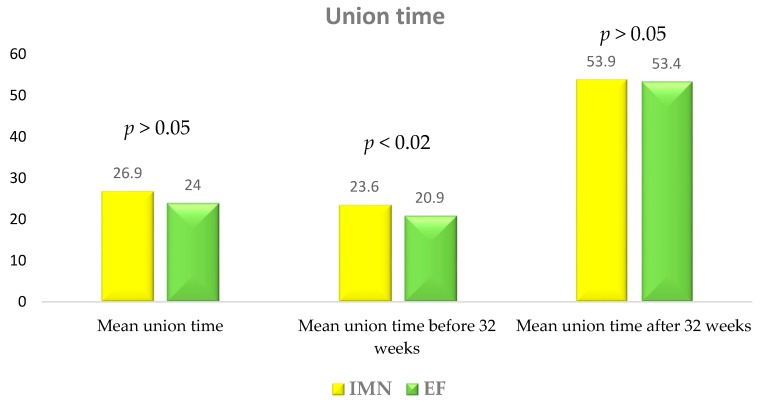
Union time: IMN versus EF group.

**Figure 3 jcm-08-01119-f003:**
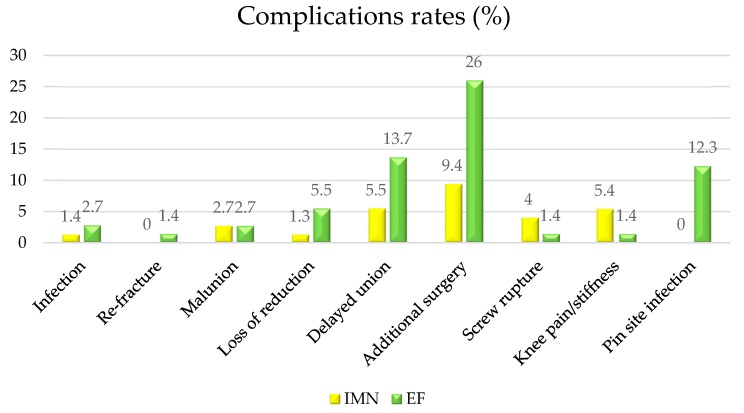
Complications: IMN versus EF group.

**Table 1 jcm-08-01119-t001:** Patients and methods: intramedullary nailing (IMN) versus external fraction (EF) group.

	IMN Group	EF Group
**Patients**	74	73
**Males**	49 (66.2%)	53 (72.6%)
**Females**	25 (33.8%)	20 (27.4%)
**Sex ratio**	2.04	2.65
**Right side**	41 (55.4%)	32 (43.8%)
**Age (years)**	42.2 ± 20.3	40.2 ± 16.5
**Polytrauma**	3 (4.1%)	12 (16.4%)

**Table 2 jcm-08-01119-t002:** Surgical timing: IMN group versus EF group.

	IMN Group	EF Group	
**Final treatment within 24 hours (pts)**	1 (1.4%)	56 (76.7%)	
**Temporary external fixation (pts)**	2 (2.7%)	12 (16.4%)	
**Delayed treatment (pts)**	71 (95.9%)	17 (23.3%)	
**Average time from fracture to surgery (days)**	5.6 ± 2.8 (range 2–20)	7.3 ± 6.1 (range 3–20)	*p* < 0.05
**Trans-skeletal traction**	71 (91%)	5 (6.8%)	
**Average surgery time (minutes)**	79.7 ± 21.7 (range 45–130)	53.8 ± 10.0 (range 35–80)	*p* < 0.0001
**Average time before post-operative weight-bearing (days)**	21.2 ± 7.4 (range 15–45)	21.8 ± 28.3 (range 0–20)	*p* > 0.05

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
