# Peer review of "Definitive Treatment of Femoral Shaft Fractures: Comparison between Anterograde Intramedullary Nailing and Monoaxial External Fixation"

_jcm, 2019, doi:10.3390/jcm8081119_

Reviewer 1 Report

This is a very interesting study. The purpose of the study is to compare two different options for femoral shaft fractures. Anterograde (IMN) and monoaxial (EF) were explored for the treatment. The authors did find that IM is the gold standard for definitive treatment of femoral shaft fractures. In patients with severe associated injuries, EF can be a good  alternative.

Overall, it is a well documented study with good presentation of results. Minor spell checks and grammar should be checked. 

Author Response

Thank you for your evaluation. We are very proud that you liked this manuscript. As you suggested a new grammar check has been performed.

Reviewer 2 Report

The information based on long-term clinical observation and survey is important. Thus, the data shown by this manuscript is good to be published. 

Here is my opinion about the modification needed: 

 The 1st paragraph (Page 5, line 148-157) and part of the 2nd paragraph (Page 5, line 158-172) in Discussion session should belong to the Introduction.

Texts were overlapped in the figure 2. Please re-format it.

Page 6, line 183-184. As it was stated as “in literature”, please add the citation.

Please add some discussion about the SF-36 score for the two groups.

Author Response

Dear Reviewer, thank you for revising our manuscript. All the suggested changes have been made.

Q1) The 1st paragraph (Page 5, line 148-157) and part of the 2nd paragraph (Page 5, line 158-172) in Discussion session should belong to the Introduction.

A1) The suggested changes were made.

Q2) Texts were overlapped in the figure 2. Please re-format it.

A2) The figure 2 was correctly formatted.

Q3) Page 6, line 183-184. As it was stated as “in literature”, please add the citation.

A3) The citations were added.

Q4) Please add some discussion about the SF-36 score for the two groups.

A4) A sentence about SF-36 score has been added in the Discussion